# The Influence of the Air Cargo Network on the Regional Economy under the Impact of High-Speed Rail in China

**Lulu Hao [1], Na Zhang [1,\*], Hongchang Li [1,\*], Jack Strauss [2], Xuejie Liu [1] and Xuemeng Guo [1]**

[1] School of Economics and Management, Beijing Jiaotong University, Beijing 100044, China; 17113113@bjtu.edu.cn (L.H.); 16113098@bjtu.eud.cn (X.L.); xmguo@bjtu.edu.cn (X.G.)

[2] Reiman School of Finance, University of Denver, 2101 S. University Blvd, Denver, CO 80208, USA; jack.strauss@du.edu

\* Correspondence: zhangna@bjtu.edu.cn (N.Z.); hchli@bjtu.edu.cn (H.L.); Tel.: +86-1890-183-6509 (H.L.)

**Abstract:** There is little research on the impact of air cargo networks on regional economic development, which is especially notable considering that Chinese airlines gradually adjusted their networks after the introduction of high-speed rail (HSR). This empirical study aims to fill this research gap. Firstly, we used the Ordinary Least Squares (OLS) method to study the effect of the air cargo network on the regional economy. The results show that, in eastern and central China, the higher the clustering coefficient of the domestic air cargo network, the more significant their promotion effect becomes on the GDP per capita, with cities in eastern China benefitting the most from this effect. However, for super-scale cities, the clustering coefficient of the domestic air cargo network has a significant negative effect on the GDP per capita, which is likely because both the air and HSR passenger services crowd out the development opportunities for air cargo. Secondly, we applied the Difference-in-Difference (DID) method in order to measure the impact on the regional economy caused by air cargo under the impact of HSR. The results show that the aviation network adjusted for the impact of HSR produces heterogeneous effects on cities for different regions and scales, and that the international aviation network has greater impacts on cities than the domestic network. In eastern China, HSR and air cargo (both international and domestic networks) promote economic growth simultaneously; in central China, only domestic air cargo has a positive effect on the regional economy; in western China, neither HSR nor air cargo has an obvious effect on the regional economy. Policy implications—such as encouraging the cooperation of HSR and civil aviation—are discussed, and could help bring the functions of the air cargo network in regional economic development into full play.

**Keywords:** high-speed rail; air cargo network; regional economy

## 1. Introduction

Experiences in China and abroad have shown that high-speed rail (HSR) can have significant advantages in delivering massive passenger volumes, shortening travel time, and improving accessibility, achieving the 'urban integration' development of different layers of cities, addressing environmental protection issues through greenhouse gas emissions reductions, and promoting the formation of a regionally integrated transportation system [1–5]. In June 2016, the Chinese State Council approved the 'Medium- and Long-Term Railway Network Plan', in which it was planned that China would construct a nationwide HSR network of 'eight vertical and eight horizontal' lines (as shown in Figure 1 below). By the end of 2019, the operating mileage of HSR reached 35,000 km in China.

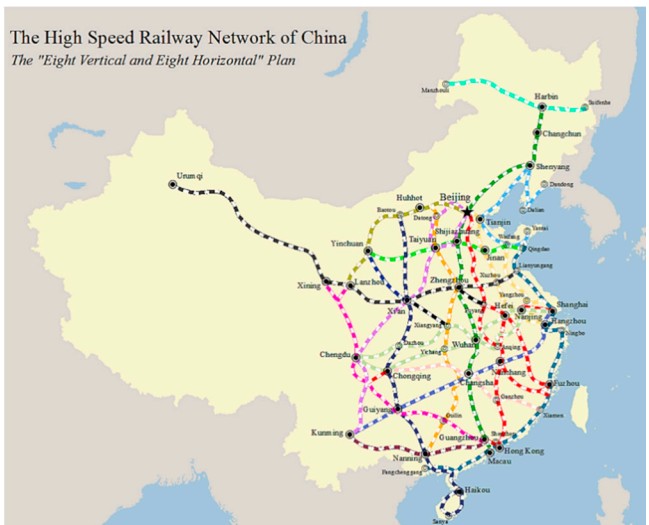

**Figure 1.** Planning map of the medium- and long-term high-speed rail (HSR) network.

The rapid development of HSR had prominent impacts on civil aviation. HSR has had a significant impact on airfares, profits, and social welfare in both the long and short term [6–9]. For example, after the opening of the Beijing–Shanghai HSR, the civil aviation passenger load factor reduced by 4%, and the ticket price dropped by 15–20%. In September 1981, the Paris–Lyon HSR was launched, which directly resulted in a 24% decrease in civil aviation passenger volume on this route [10]. In November 1994, Eurostar was officially put into operation. One year later, the London–Paris air passenger volume decreased by 70%, and that for the London–Brussels route decreased by 65% [11]. After Japan's Shinkansen was put into operation, within traveling distances of 200 to 800 km from major stations, air and road passenger volumes both decreased by 35% [12].

According to transportation economics, air transportation that shortens the space and time distances among cities can not only optimize labor division structures but can also produce regional economic benefits [13]. China's civil aviation development strategy focuses predominantly on passenger, rather than freight, transportation [14]. However, Air Transport Action Group (ATAG) research has shown that, although air transports less than 1% of the total global traffic volume of goods, it provides a value contribution of 35% for total international trade, and more than 4.5% of global economic output can be attributed to air transport. Thus, air cargo transport is an important way for governments to boost their economic and social development [15,16]. The air cargo network determines, to some extent, the accessibility, stability, reliability, and efficiency of air cargo transportation, and provides new potential opportunities for regional economic development [17]. Therefore, this study aims to explore, under the influence of HSR and with the corresponding adjustment of the air cargo network, the impacts of air cargo on the regional economy of China.

## 2. Literature Review and Analytical Framework

### 2.1. Literature Review

#### 2.1.1. HSR and Air Transport

The rapid development of HSR in China is causing unprecedented challenges for the civil aviation industry [18]. Generally speaking, HSR has advantages over civil aviation in terms of travel time, ticket prices, traffic frequency, accessibility to HSR stations, and punctuality, etc.; thus, it has considerable impacts on the civil aviation industry [19,20]. Tan (2011) [21] systematically studied intermodal competition between HSR and civil aviation, covering five factors: ticket prices, time value, subjective passenger utility, objective safety, and punctuality. Tan maintained that the opening of HSR will inevitably affect the development of China's civil aviation transportation industry, while the magnitude of these impacts

could differ considerably over time. Yonghwa and Hun (2006) [22] used the Seoul–Daegu HSR section in South Korea as an example in order to study the changes in the passenger demand for civil aviation after HSR became available. The authors constructed a utility function equation for the civil aviation passengers, with travel time, ticket price, and train frequency as variables before and after the introduction of HSR, and collected data through a questionnaire survey in order to estimate the relevant parameters. Their research results showed that, after HSR became available, only 28% of civil aviation passengers continued to choose airlines for travel, while most passengers transfer to HSR. Roman et al. (2007) [23] analyzed the competitive relationship between HSR and airlines in the Madrid–Barcelona transportation corridor in Spain using a mixed revealed preferences (RP)/stated preferences (SP) database in order to estimate the parameters in the discrete choice model, and then constructed a willingness-to-pay function for the different passenger groups in order to reveal the prices they were willing to pay for HSR's improved service quality. The authors found that HSR has comparative competitiveness to civil aviation for medium- and long-distance market segments, largely because HSR reduces travel hours and increases train punctuality. Zhang, Luan, and Cai (2011) [24] built a competition model between HSR and civil aviation by applying a Logit model. Based on the operational data of HSR and civil aviation, the authors found that the intensity of competition between HSR and civil aviation changes as the travel distance varies. The authors also analyzed the intermodal competition outcomes for the affected civil aviation routes. Mao, Li, and Zhou (1996) [25] developed a qualitative study of the market share among car, railway (including HSR), air, and bus travel in European countries, and found that the competitive distance range for HSR is approximately 200 to 1200 km, while air transportation has a dominant advantage for distances over 1500 km.

In general, the operation of HSR has impacts on civil aviation transportation; in particular, it has a strong substitution effect on the civil aviation routes that overlap with those of HSR [26]. After Japan's Shinkansen began operating, for travel distances that ranged from 200 to 800 km, the railway market share increased dramatically from 30% to 65%, diverting a large volume of air passengers [12]. After the Paris–Lyon HSR began operating, 24% of civil aviation passengers shifted to HSR and forced Air France to exit from the Paris–Lyon and Paris–Marseille civil aviation markets [10]. One year after the opening of the Eurostar HSR, this HSR quickly became the travel preference for 70% of the total passengers traveling between the UK and Paris, and, as a consequence, the airlines canceled most of their air services from London to Paris and Brussels [11]. The gradual construction of China's 'eight vertical and eight horizontal' HSR network lines resulted in the successive termination of the Zhengzhou–Xi'an, Tianjin–Nanjing, and Beijing–Xuzhou civil aviation routes.

### 2.1.2. Air Transport and Regional Economy

Scholars at home and abroad have carried out considerable research on the impacts of the development of air transport on the regional economy. The development of air transportation can drive the development in areas surrounding airports. With airports at the core, civil aviation-related industries cluster, new civil aviation metropolitan areas form, and civil aviation transport corridors and industrial belts expand. Therefore, civil aviation plays an important role in promoting the regional economy [27]. Meanwhile, due to its location advantages and the aggregation of advanced production factors, civil aviation also has a strong spillover effect on the regional economy. Developing the civil aviation economy is an important way to promote regional economic growth, narrow regional income gaps, and achieve coordinated regional economic development [28]. Huddleston and Pangotra (1990) [29] argued that a close relationship exists between air transport and the regional economy because air transport can significantly improve the accessibility between different regions, increase regional incomes, and boost employment. Bruckner (2006) [30] maintained that the air passenger and cargo transportation service systems of airports are key factors to determine the effect and degree of the impacts of civil aviation on the urban economy. After a large number of field investigations, Kasarda and Jonathan (2005) [31] found that the civil aviation cargo transportation industry has a great impact on international trade, and is an engine that stimulates urban economic development. Keith (1999) [32]

studied multiple airports in the Carolinas, USA, and found that the regional employment level increases with an increase in passenger and cargo throughput, and, under the influence of the civil aviation industry, the regional industrial structure gradually evolves to become a high-tech one. David and Ashish (2002) [33] concluded that the surrounding areas of airports are advantageous geographic locations for regional economic development, and that they can serve as important distribution centers for regional passengers, goods, information, business, and capital flows. Gradually, various modern social and economic activities agglomerate, and brand-new urban forms come into being.

The air transportation industry provides services through its network systems (airline networks, airport networks, air traffic control networks, and reservation system networks, etc.). The network externalities of air transportation enable airports to generate significant spatial spillover effects. In other words, the relationship between air transportation and the regional economy is a complex mechanism defining an air transport network's impact on the overall economic system [34]. Air routes are the core resources for airlines, and are also the basis of competitiveness and development for the airports and the aviation industry [35]. China's air cargo network is widely distributed, with an average path length of 2.26 and a clustering coefficient of 0.76. Moreover, its subsection fitting power indexes are 0.398 and 2.141 [18]. These characteristics demonstrate that China's air cargo network is a scale-free network [36]. Notably, China's air cargo transport hubs strongly overlap with its passenger transport hubs. At present, the city-to-city route network structure still prevails in China's domestic air cargo network, with features of imbalance and multiple structures [10]. Studies on the relationship between the civil aviation network and regional economic development show that airports, especially major hubs, play roles in promoting regional economic development, mainly in the following ways: by providing the cities and regions with access to the air transport system; by creating local employment opportunities; and by promoting other related businesses, such as the development of auxiliary services inside or outside airports [37].

## 2.2. Analytical Framework

Experiences in China and beyond show that, with the expansion of HSR, civil aviation gradually loses its traditional market segments. HSR even erodes some long-distance civil aviation transportation markets. In particular, the average haul distance for air cargo continues to expand under the impact of HSR. Our analytical framework is shown in Figure 2.

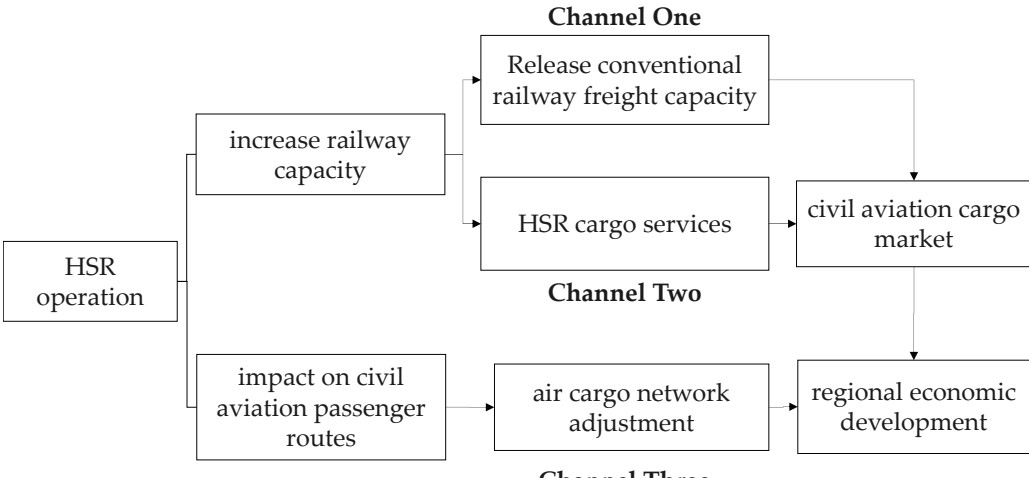

**Figure 2.** Analytical framework for the impacts of HSR and civil aviation on the regional economy.

According to Figure 2, there are three specific ways for HSR to produce impacts on civil aviation cargo services and the subsequent regional economic development.

### 2.2.1. Channel One: Conventional Railway Capacity Release and Regional Economic Development

The operation of the HSR shifts passengers' demand away from conventional railways to HSR, and relieves the transport capacity on conventional lines, making them available for the freight market. China's HSR has been developing into a network covering more than 80% of the large cities located in the regions of the Yangtze River Delta, the Bohai Rim, and the Pearl River Delta. A large proportion of conventional railway passengers transfer to HSR, which effectively alleviates the conventional railway's freight bottleneck. For example, at the beginning of the Beijing–Shanghai HSR's operations, the corresponding cargo capacity on conventional railways increased by around 50 million tons. The Ministry of Transport and the China Railway Group promoted and implemented a strategy of passenger–freight separation, increasing investments in railway freight facilities, developing multi-modal transportation to organically connect highways, developing water and railway infrastructure, and setting up a coordination mechanism for intermodal operation and management to increase the loading and reloading efficiency, which contributes to the expansion of railway freight capacity and the market attraction towards this form of expansion. Finally, HSR and the corresponding freight release on conventional railways will have an impact on regional economic development.

### 2.2.2. Channel Two: HSR Cargo Service and Regional Economic Development

HSR in China now provides express freight services for society, with an obvious advantage over civil aviation, given its massive volumes, punctuality, timeliness, and comparatively low prices. By the end of 2018, HSR express services in China covered 44 cities and 161 HSR lines. There are two main types of air freight services with which HSR can compete: services for air cargo with high value, and services for air cargo demands with high time sensitivity; in other words, these types of air cargo have to be transported in a rapid and urgent way. Traditionally, there is little substitutability between conventional railway and civil aviation services because they provide the market with very different freight services in terms of cargo quantity, volume, time, and value, etc. For example, conventional railways are regarded as being suitable for the transportation of massive goods, while civil aviation is better for long-distance and high-value-added cargo transportation. However, the opening of HSR changed the competitive landscape between HSR and civil aviation in the freight market. HSR can compete with civil aviation for long-distance and high-value-added cargo, given its flexibility and punctuality (in China, the punctuality for airlines is lower than that of HSR). The on-time rate of HSR trains is 98.8% (Data source: National Railway Administration of the People's Republic of China, website: http://www.nra.gov.cn/), and that for air is 71.25% (Data source: Civil Aviation in Statistical Yearbook, 2008–2017). Therefore, HSR will change China's domestic freight transportation market structure and replace some air cargo services. Thus, together with air cargo services, HSR will have certain impacts on regional economic development.

### 2.2.3. Channel Three: Air Cargo Network Adjustment and Regional Economic Development

To deal with the fierce competition from HSR, Chinese airlines are forced to adjust their domestic and international fight routes and even withdraw completely from some regional civil aviation market segments. In particular, in the overlapping market of short distance HSR and civil aviation in China, the airlines canceled almost all their fight services. For example, in April 2009, the opening of the Shijiazhuang–Taiyuan HSR directly led to the cancellation of all of the morning and evening flights between the two cities. In November 2009, the Chengdu–Chongqing HSR was put into operation; subsequently, the flight seat load factor between Chengdu and Chongqing dropped to less than 50%, which means that the airlines suffered losses. As a result, all of the flights between the two cities were canceled. In February 2010, with the opening of the Zhengzhou–Xi'an HSR, airlines including Happiness Airlines, Kunpeng Airlines, and Spring Airlines stopped their intercity air flight services successively. Driven by HSR competition, Chinese airlines not only adjusted their aircraft types,

flight frequencies, and domestic and international fight route structures but also increased their number of medium and long distance flights, which eventually resulted in the development of a new civil aviation network. For a long period of time, the Chinese civil aviation industry adopted a passenger transport priority strategy, and preferentially allocated transportation resources to the development of air passenger transport, rather than air cargo business; that is, they paid "heavy attention to air passenger transport, but little attention to cargo transport, and air cargo transport following air passenger transport". The air cargo transport mode is called the 'passenger aircraft belly transportation mode', which means that air cargo service providers mainly take advantage of the belly cargo capacity of passenger aircrafts in order to deliver cargo services to the market [38]. Therefore, the direct impact of HSR on air passenger services has an indirect influence on air cargo services and thus regional economic development. Moreover, insufficient investments in air cargo facilities also cause the competitiveness of air cargo to decline and hinder air cargo's contribution to regional economic development.

To summarize, the operation of China's HSR network not only relieves the freight transportation capacity on conventional railways by transferring passengers from conventional railways to HSR but also has direct impacts on short- and medium-distance air passenger transportation, and indirect impacts on the air cargo network. Meanwhile, HSR can also compete with air cargo businesses by providing the market with express HSR cargo services. Hence, HSR, civil aviation and the interaction between HSR and civil aviation will influence regional economic development.

## 3. The Model

### 3.1. Model Construction

#### 3.1.1. Air Cargo Network and Regional Economy

The Ordinary Least Square (OLS) method is widely used to study the impact of the transportation infrastructure on economic development [39,40]. As one of the most fundamental forms of regression analysis, the OLS method is based on the basic principle that the optimal fitting curve should be the smallest sum of the squares (i.e., the sum of the squares of the residual) of the distance from each point to the line. In this section, Equation (1) is introduced in order to study the impact of air freight on the regional economy by using the OLS method. In the next section, based on Equation (1), we then apply the DID model in order to study the impact of air freight on the regional economy under the impact of HSR.

$$Y_{it} = \alpha_0 + \alpha_1 C_{it} + \alpha_2 FDI_{it} + \alpha_3 SEC_{it} + \alpha_4 LAB_{it} + \alpha_5 K_{it} + \alpha_6 R\&D_{it} + \varepsilon_{it} \tag{1}$$

where the dependent variable $Y_{it}$ is the GDP per capita for city $i$ at year $t$; $\alpha_0$ is a constant term; $\alpha_i$ ($i = 1, 2, 3, 4, 5, 6$) are the regression coefficients of the controlled variables; $C_{it}$ is the clustering coefficient of the air cargo network for the airport of city $i$ in year $t$ (for each airport for a specific city, we calculate the corresponding air cargo network clustering coefficient (see Equation (5)); $FDI_{it}$ is the foreign direct investment for city $i$ at year $t$; $SEC_{it}$ is the proportion of the secondary industry production value to the overall GDP for city $i$ at year $t$; $LAB_{it}$ is the number of employees for city $i$ at year $t$; $K_{it}$ is the real fixed asset investment for city $i$ at year $t$; $R\&D_{it}$ is the research and development expenditures for city $i$ at year $t$; and $\varepsilon_{it}$ is the error term.

#### 3.1.2. HSR, Air Cargo Network and Regional Economy

In this section, we explore the effects and the magnitude of the air cargo network clustering coefficient for different airports on regional economic development under the impact of HSR. The difficulty of the policy effect evaluation lies in the endogenous nature of economic events and economic policy [41], which also applies to our research. In order to overcome the endogeneity problem, economists often use quasi-experiments and various econometric tools to evaluate the effects of economic events or policies. Popular models include, but are not limited to,

the instrumental variable method, breakpoint regression, the propensity score matching method, and the Differences-in-Differences (DID) method [42].

In the late 1970s, scholars began to introduce the DID method into the economic research field in order to study economic problems [43]. In China, to the best of our knowledge, Zhou and Chen (2005) were the first to use the DID method to evaluate public policy effects [44], focusing on the impact of taxation reform policy on farmers' income growth. The basic idea of the DID method is to treat public policy or an economic event as a natural experiment. In order to evaluate the net impact of an economic policy or event, all of the sample data are divided into two groups: one group is affected by the policy or event, i.e., the treatment group; the other is not affected by the policy or event, i.e., the control group. After the first differences for the economic variables, the heterogeneity of the variables that do not change over time can be eliminated, and then the second difference is measured between the two sets of variables in order to eliminate the incremental effects that change over time, thus obtaining the net effect of the policy or event. The basic form of the DID model is as follows:

$$Y_{it} = \beta_0 + \beta_1 Treated_i + \beta_2 Period_t + \beta_3 Treated_i \times Period_t + \delta_{it} \tag{2}$$

where $Y_{it}$ is the same meaning as in Equation (1); $Treated_i$ is a grouping variable, with 1 indicating the treatment group and 0 indicating the control group; $Period_t$ is a time variable, with 1 indicating the case after the policy or event occurs and 0 indicating the case before the policy or event occurs for city $i$; the interaction term $Treated_i \times Period_t$ is the effect of the treatment group after the HSR becomes available; $\beta_0$ is a constant term; $\beta_1$ and $\beta_2$ are the regression coefficients for the grouping variable and the time variable, respectively; $\beta_3$ is the coefficient of the interaction term; and $\delta_{it}$ is the error term. However, Equation (2) assumes that, for the cities in the treatment group, HSRs are put into operation simultaneously, which is not consistent with the fact of the gradual expansion of the HSR network. Therefore, based on Equation (2), we use the multi-period DID method to define cities that have no HSR for the control group, and cities that have HSR as the treatment group. Furthermore, the city fixed effects and time effects are controlled. The multi-period DID model is shown in Equation (3), as follows:

$$Y_{it} = \beta_0 + \beta_3 Treated_{it} \times Period_{it} + \eta_{it} + \theta_{it} + \gamma X_{it} + \delta_{it} \tag{3}$$

where $Y_{it}$ is the same as in Equations (1) and (2); $\eta_{it}$ is the fixed effect for city $i$ at year $t$; $\theta_{it}$ is the fixed effect of time for city $i$ at year $t$; $X_{it}$ is the other variables controlled for city $i$ at year $t$; $\gamma$ is the coefficient matrix for the controlled variables; $\beta_0$ is a constant term; $\beta_3$ is the coefficient of the interaction term for city $i$ at year $t$; and $\delta_{it}$ is the error term.

In addition, this research focuses not only on the impact of the HSR on the regional economy but also on the influence of the air cargo network for different airports located in different cities on regional economic development after the introduction of HSR. Therefore, based on Equations (1) and (3), above, the final model is shown in Equation (4), as follows:

$$\begin{aligned} Y_{it} = {} & \beta_0 + \beta_3 Treated_{it} \times Period_{it} + \beta_4 Treated_{it} \times Period_{it} \times C_{it} + \eta_{it} + \theta_{it} \\ & + \gamma_1 FDI_{it} + \gamma_2 SEC_{it} + \gamma_3 LAB_{it} + \gamma_4 K_{it} + \gamma_5 R\&D_{it} + \delta_{it} \end{aligned} \tag{4}$$

where $Treated_{it} \times Period_{it} \times C_{it}$ is the change effect of the air cargo network after HSR enters city $i$ at year $t$, and the coefficient $\beta_4$ measures the impact of the air cargo network clustering coefficient for a specific airport located at city $i$ at year $t$ on the regional economy. Notably, $C_{it}$ is removed from Equation (4) for collinearity considerations. The other terms are the same as in Equations (1)–(3).

## 3.2. Variable Definitions and Sensitivity Analysis

### 3.2.1. Variable Definitions

Based on the data availability, the variables selected and their corresponding explanations are illustrated as follows.

(1)　$Y_{it}$: in order to measure the development of the regional economy, in reference to the common practices in Chinese and non-Chinese research, this study uses the real GDP per capita for city $i$ at year $t$.

(2)　$Treated_{it}$: the availability of HSR is set as a dummy variable for city $i$ at year $t$. The data period is from 2008 to 2017. A value of 1 or 0 is assigned as the dummy variable of HSR availability during the study period. The cities with HSR are classified under the treatment group, and the assigned value is set as 1; cities without HSR fall under the control group, and the assigned value is set as 0.

(3)　$Period_{it}$: for city $i$ at year $t$, when the HSR is put into operation, the years after the HSR's opening are set as 1 and 0 otherwise.

(4)　$FDI_{it}$: the actual amount of foreign investments in city $i$ at year $t$, which measures the level of trade development in the city concerned.

(5)　$SEC_{it}$: the proportion of the secondary industry production value to the overall GDP for city $i$ at year $t$. This variable indicates that industrial structure and freight transportation are closely related to the development of secondary industry, which is the main source of the derived freight demand.

(6)　$LAB_{it}$: the number of employees for city $i$ at year $t$, which can measure the actual human resources engaged in production and business activities.

(7)　$K_{it}$ : the actual fixed asset investment for city $i$ at year $t$, which can measure the level of city investment.

(8)　$R\&D_{it}$: the direct research and development investment for city $i$ at year $t$, which can measure the technological inputs.

(9)　$C_{it}$: this variable measures the clustering coefficient of the air cargo network for the airport of city $i$ at year $t$. In our study, all of the Chinese airports, particularly airports with air cargo exchanges during the 10 year period from 2008 to 2017, are considered (as shown in Table 1). Each airport's clustering coefficient is calculated to reflect its air cargo network characteristics. In other words, each airport is regarded as a hub for its air cargo network.

**Table 1.** Statistical description of the airports and air cargo routes involved.

| Year | Domestic | | | International | | |
|---|---|---|---|---|---|---|
| | Total Airport Number | Number of Airports with Air Cargo Activity | Domestic Air Cargo Routes Number | Total Airport Number | Number of Airports with Air Cargo Activity | International Air Cargo Route Number |
| 2008 | 150 | 67 | 961 | 104 | 93 | 6027 |
| 2009 | 163 | 72 | 1094 | 93 | 87 | 4644 |
| 2010 | 172 | 75 | 1211 | 110 | 90 | 5185 |
| 2011 | 175 | 62 | 1121 | 126 | 114 | 6147 |
| 2012 | 178 | 63 | 1063 | 121 | 118 | 7035 |
| 2013 | 188 | 64 | 905 | 118 | 118 | 7281 |
| 2014 | 198 | 50 | 540 | 123 | 121 | 7482 |
| 2015 | 204 | 52 | 475 | 137 | 126 | 7739 |
| 2016 | 212 | 57 | 663 | 146 | 137 | 8199 |
| 2017 | 224 | 56 | 737 | 158 | 151 | 8840 |

In order to measure the air cargo network for different airports in China, this study applies complex network theory and methods. Wattsk and Strogtz (1998) [45] argue that a type of network exists that is different from a regular network and a random network: a small world network. This network is between a regular network and a random network; its clustering coefficient is larger than that of a random network, and is similar to a regular network; and its average path length is much smaller than that of a regular network, and is similar to a random network. The civil aviation network has the typical characteristics of a small-world network [46–48]. The clustering coefficient (also called the agglomeration coefficient) of the air cargo network for a specific airport can reflect the hierarchical

nature of the airport as being important to the overall air cargo business. The clustering coefficient for airport *i* can be expressed by Equation (5), as follows:

$$C_{it} = \frac{E_{it}}{k_{it}(k_{it}-1)/2}, \ (k_{it} \geq 2) \tag{5}$$

where $C_{it}$ is the clustering coefficient for airport *i* (that is, city *i*, since airport *i* is located at city *i*) in year *t*. $E_{it}$ is the number of the real cargo air route connections (i.e., the real connected edges) between airport *i* at year *t* and other airports. The number of connections is valid only if there is actual air cargo transport between city *i* and other airports at year *t*. For example, for the year 2017, 39 domestic airports engaged in air cargo transportation activities with the Beijing International Airport, so $E_{Beijing,2017}$ is equal to 39. Variable $k_{it}$ is the total number of all of the possible connections (i.e., the maximum possible edges in theory) between airport *i* and the other airports at year *t*. For example, for 2017, there are 224 domestic airports, including Beijing International Airport, that engage in air cargo transport activities, so $k_{Beijing,2017}$ is equal to 224. The denominator for Equation (5) is the total number of all of the possible connections [49] between airport *i* (that is, city *i*) and other airports at year *t*. A larger value of $C_{it}$ indicates that airport *i* is closely linked to other airports in providing air cargo services, and vice versa.

Based on Equation (5), we measure the clustering coefficient of the domestic air cargo network ($C_{1it}$) and the clustering coefficient of the international air cargo network ($C_{2it}$) for each city *i*'s airport at year *t*. When the clustering coefficient of the domestic air cargo network ($C_{1it}$) is calculated, $E_{it}$ considers all of the Chinese airports connected to airport *i* at year *t*. When the clustering coefficient of the international air cargo network ($C_{2it}$) is calculated, $E_{it}$ considers all of the foreign airports connected to airport *i* at year *t*. For example, in 2017, the number of international airports that actually engaged in air cargo business with Beijing International Airport was 114 (Data source: General Administration of Customs of China, website: http://www.customs.gov.cn/); that is, $E_{it}$ is equal to 114, while the number of total airports that can connect to Beijing International Airport is 158. Thus, $k_{it}$ is equal to 158 (Data source: Civil Aviation in Statistical Yearbook, 2008–2017).

We follow the related studies in which an undirected network is often used to study the impact of HSR or a civil aviation network on regional economic development. In particular, researchers prefer to adopt parameters which can reveal the relative importance of an individual airport, such as clustering or the agglomeration coefficient [50–52], when the relationship between civil aviation and the regional economy is assessed.

### 3.2.2. Sensitivity Analysis

There are abundant studies on economic growth. Levine and Renelt (1992) [53] argued that only a few variables can have impacts on economic growth. The authors found that the GDP per capita, the proportion of investments to GDP, and the enrollment rate of junior high school students had stable and significant influences on economic growth. Sala-i-martin (1997) [54] maintained that the standards of Levine and Renelt (1992) were too strict. After lowering them, he found that the rate of inflation, the degree of property rights protection, and the efficiency of the capital market could also significantly affect economic growth. In this paper, a sensitivity analysis was performed based on the method developed by Chen and Kerk (2005) [55] in order to assess the sensitivity of variables on the regional economy (as shown in Table 2).

**Table 2.** The results of the sensitivity analysis.

| Variable | The Proportion of Times When the Regression Coefficient is Significant | | | Significant |
|:---:|:---:|:---:|:---:|:---:|
| | **90%** | **95%** | **99%** | |
| $C_1$ | 0% | 50% | 33% | ** |
| $C_2$ | 17% | 32% | 0% | * |
| FDI | 13% | 0% | 0% | - |
| SEC | 17% | 25% | 25% | ** |
| LAB | 0% | 0% | 92% | *** |
| K | 0% | 0% | 67% | *** |
| R&D | 11% | 0% | 89% | *** |

Note: * stands for significant at the 10% level, ** stands for significant at the 5% level, and *** stands for significant at the 1% level. It can be seen from Table 2 that all of the variables selected in this paper have significant impacts on regional economic growth. The detailed analysis is presented in Section 4.

*3.3. The Data*

3.3.1. Data Sources

The data for the GDP per capita, foreign direct investments, the proportion of secondary industry production value to the overall GDP, employees, the real fixed asset investments, and R&D expenditures are from the China City Statistical Yearbook from 2008 to 2017. The data required to calculate the air cargo network clustering coefficient are from the Civil Aviation section in the Statistical Yearbook, from 2008 to 2017.

3.3.2. Description of Data

The variables used in the model are described in Table 3.

**Table 3.** Description of the data.

| Variable | Obs. | Mean | S.d. | Min. | Max. |
|:---:|:---:|:---:|:---:|:---:|:---:|
| $C_1$ | 760 | 0.00072 | 0.00096 | 0 | 0.00733 |
| $C_2$ | 760 | 0.01209 | 0.01508 | 0 | 0.0748 |
| FDI | 760 | 0.02792 | 0.03722 | 0.00004 | 0.78891 |
| SEC | 760 | 45.98 | 11.21 | 14.95 | 85.08 |
| LAB | 760 | 434,162 | 1,096,637 | 94,980 | 9,868,700 |
| K | 760 | 217,000 | 214,000 | 10,579 | 1,720,000 |
| R&D | 760 | 1,132,261 | 1,683,187 | 13,277 | 13,300,000 |

## 4. Empirical Results

In this paper, the OLS and DID methods are applied in order to study the impacts of HSR, the air cargo network, and their interactions on the regional economic development at three scales: all of China, different regions (eastern, central, and western China), and cities of different sizes.

*4.1. Empirical Results with the OLS Method*

4.1.1. Impacts of the Air Cargo Network on All of China and Different Regions

By using the OLS regression model, we explore the impact of China's air cargo network on the regional economy (HSR is excluded), as shown in Table 4.

**Table 4.** Regression results for cities in different regions with the Ordinary Least Squares (OLS) method.

| Variable | Overall | | | Eastern China | | | Central China | | | Western China | | |
|---|---|---|---|---|---|---|---|---|---|---|---|---|
| | (1) | (2) | (3) | (4) | (5) | (6) | (7) | (8) | (9) | (10) | (11) | (12) |
| $C_1$ | 0.030 | 0.029 | | 0.033 | 0.032 | | 0.026 | 0.024 | | 0.018 | 0.017 | |
| | (6.51) *** | (6.34) *** | | (6.11) *** | (6.08) *** | | (3.35) ** | (3.08) ** | | (2.06) | (1.95) | |
| $C_2$ | 0.015 | | 0.012 | 0.006 | | 0.002 | 0.019 | | 0.017 | 0.016 | | 0.015 |
| | (2.28) | | (1.78) | (0.75) | | (0.28) | (2.39) | | (1.99) | (1.40) | | (1.24) |
| FDI | −0.033 | −0.031 | −0.020 | −0.014 | −0.014 | 0.004 | −0.028 | −0.028 | −0.015 | −0.050 | −0.045 | −0.046 |
| | (−4.79) *** | (−4.52) *** | (−2.89) ** | (−1.53) | (−1.49) | (0.46) | (−2.17) | (−2.17) | (−1.20) | (−4.13) *** | (−3.87) *** | (−3.78) ** |
| SEC | −0.050 | −0.051 | −0.065 | −0.067 | −0.066 | −0.089 | 0.004 | 0.007 | 0.002 | −0.019 | −0.025 | −0.025 |
| | (−3.58) *** | (−3.67) *** | (−4.62) *** | (−3.55) *** | (−3.54) *** | (−4.60) *** | (0.18) | (0.30) | (0.07) | (−0.79) | (−1.01) | (−0.99) |
| LAB | −0.019 | −0.020 | −0.015 | −0.026 | −0.024 | −0.012 | 0.001 | 0.011 | −0.003 | −0.024 | −0.027 | −0.026 |
| | (−1.39) | (−1.39) | (−1.06) | (−1.12) | (−1.04) | (−0.49) | (0.04) | (0.33) | (−0.07) | (−1.30) | (−1.50) | (−1.39) |
| K | −0.107 | −0.110 | −0.124 | −0.899 | −0.909 | −1.032 | −0.934 | −1.054 | −1.203 | 0.069 | 0.071 | 0.070 |
| | (−1.83) | (−1.87) | (−2.04) | (−5.65) *** | (−5.74) *** | (−6.24) *** | (−4.52) *** | (−5.13) *** | (−6.00) *** | (1.05) | (1.07) | (1.05) |
| R&D | 0.204 | 0.203 | 0.214 | −0.007 | −0.008 | −0.008 | 0.832 | 0.913 | −0.035 | 0.585 | 0.585 | 0.580 |
| | (7.68) *** | (7.64) *** | (7.85) *** | (−0.54) | (−0.61) | (−0.59) | (6.86) *** | (6.56) *** | (7.29) *** | (8.63) *** | (8.61) *** | (8.49) *** |
| $R^2$ | 0.802 | 0.800 | 0.790 | 0.827 | 0.827 | 0.809 | 0.919 | 0.914 | 0.909 | 0.800 | 0.799 | 0.797 |

Note: ** stands for significant at the 5% level, and *** stands for significant at 1% level.

From the overall perspective of China, the coefficient of $C_1$ is about 0.03, and the coefficients of $C_1$ for eastern, central, and western China are about 0.033, 0.026, and 0.018, respectively. Therefore, the domestic air cargo network can promote overall economic development, and cities located in eastern China benefit the most from the perfection of the air cargo network and the improvements in its corresponding clustering coefficient. The coefficient of $C_1$ for western China is not significant for two possible reasons: the air cargo network may not be sufficiently developed to support regional economic growth, or the regional economic structure may not rely very heavily on air cargo transportation. For cities in eastern and central China, which have more flight routes to connect specific airports to provide air cargo services, the magnitude of $C_1$ will become larger, and the corresponding impact on the regional economy will also increase.

The coefficient of $C_2$ for all of China is around 0.015, while the coefficients of $C_1$ for eastern, central, and western China are around 0.006, 0.019, and 0.016, respectively. It is clear that the impact of the international air cargo network on the regional economy is much less than that of the domestic cargo network. The observation that the clustering coefficient of the international air cargo network produces insignificant impacts on China overall, and on regional economic development, somewhat contradicts our expectations. There are two possible explanations for this observation: (1) there are too few international air routes and too incomplete an air cargo service network in western China to have significant impacts on the regional economy; (2) Chinese air cargo transportation mainly relies on the belly capacity of passenger aircraft, which cannot provide sufficient, efficient, or just-in-time (JIT) services for large cities such as Beijing, Shanghai, and Guangzhou. For large countries like China or the U.S.A., with vast territories, a dedicated air cargo network is needed in order to satisfy air cargo transport demands [52]. For other control variables, FDI, SEC, and K have negative effects on the GDP per capita, R&D plays a positive role in promoting GDP per capita growth, and LAB (the number of employment) has no significant effect.

### 4.1.2. Impacts of the Air Cargo Network on Cities of Different Scales

In our study, we separated the cities into two groups: super-large cities, and cities. The OLS regression results for cities of different scales are shown in Table 5.

The coefficient of $C_1$ for super-large cities is −0.436, and that for the other cities is a positive 0.026. This provides a very sharp contrast, since the air cargo network has a negative impact with a large magnitude on the regional economic growth for super-large cities, and a positive impact on regional economic growth for other cities. The most reasonable explanation for this phenomenon is that there is a crowding effect of air passengers on air cargo for super-large cities. At present, there are still no dedicated cargo airports in China, and the airports in super-large cities mainly focus on air passenger transportation. Few time slots and other facilities are devoted to air passenger demands rather than air cargo. Thus, the crowding effect leads to the low efficiency of air cargo transport and inhibits the development of the regional economy.

The coefficient of $C_2$ for super-large cities is insignificant, and that for other cities is 0.021. For other cities, the positive coefficient of $C_2$, which is line with our expectations, indicates that the expansion of the international air cargo network can promote regional economic growth. For super-large cities, the reason is similar to the explanation of the $C_1$ for super-large cities, as shown in Table 4. Another reason is that although FedEx, UPS, and DHL established air cargo distribution centers in Guangzhou, Shenzhen, and Shanghai (which are super-large cities), respectively, these air cargo centers have not yet had an observable impact on the regional economy, possibly due to the time lag effect. In other words, those international air cargo facilities will demonstrate their impact on regional economic growth only in later years. For the control variables, FDI has a negative impact on the other cities and no impact on super-large cities, SEC has a negative impact on the GDP per capita, R&D has a positive impact on the GDP per capita for other cities but no impact on super-large cities, and LAB and K have no significant impacts.

**Table 5.** Regression results for cities of different scales with the OLS method.

| Variable | Overall | | | Super-Large Cities | | | Other Cities | | |
|---|---|---|---|---|---|---|---|---|---|
| | (1) | (2) | (3) | (4) | (5) | (6) | (7) | (8) | (9) |
| $C_1$ | 0.030 | 0.029 | | −0.436 | −0.447 | | 0.026 | 0.026 | |
| | (6.514) *** | (6.34) *** | | (−5.05) *** | (−5.41) *** | | (6.40) *** | (6.30) *** | |
| $C_2$ | 0.015 | | 0.012 | −0.063 | | −0.241 | 0.021 | | 0.021 |
| | (2.28) | | (1.78) | (−0.49) | | (−1.77) | (3.80) *** | | (3.63) *** |
| FDI | −0.033 | −0.031 | −0.020 | 0.150 | 0.155 | −0.106 | −0.028 | −0.026 | −0.016 |
| | (−4.79) *** | (−4.52) *** | (−2.89) ** | (2.26) | (2.37) | (−2.26) | (−4.21) *** | (−3.80) *** | (−2.41) |
| SEC | −0.050 | −0.051 | −0.065 | −0.244 | −0.251 | 0.058 | −0.038 | −0.038 | −0.052 |
| | (−3.58) *** | (−3.67) *** | (−4.62) *** | (−2.99) ** | (−3.14) ** | (0.95) | (−2.43) | (−2.45) | (−3.26) ** |
| LAB | −0.019 | −0.020 | −0.015 | 0.067 | 0.065 | 0.053 | −0.010 | −0.011 | −0.004 |
| | (−1.39) | (−1.39) | (−1.06) | (1.86) | (1.84) | (1.35) | (−0.79) | (−0.84) | (−0.31) |
| K | −0.107 | −0.110 | −0.124 | −0.015 | −0.029 | −0.639 | −0.017 | −0.026 | −0.032 |
| | (−1.83) | (−1.87) | (−2.04) | (−0.05) | (−0.10) | (−2.24) | (−0.31) | (−0.46) | (−0.54) |
| R&D | 0.204 | 0.203 | 0.214 | 0.040 | 0.041 | 0.048 | 0.586 | 0.583 | 0.625 |
| | (7.68) *** | (7.64) *** | (7.85) *** | (1.68) | (1.73) | (1.84) | (11.58) *** | (11.40) *** | (12.03) *** |
| $R_2$ | 0.802 | 0.800 | 0.790 | 0.852 | 0.852 | 0.818 | 0.834 | 0.829 | 0.821 |

Note: ** stands for significant at the 5% level, and *** stands for significant at the 1% level.



*4.2. Empirical Results with the DID Method*

Tables 3 and 4 illustrate the impact of $C_1$ and $C_2$ on the regional economy, while Tables 5 and 6 demonstrate the different levels of impact after the introduction of HSR.

4.2.1. Impacts of the Air Cargo Network on China Overall and the Different Regions

In Equation (4), the DID model is used to study the impact of the air cargo network on the regional economy under the influence of HSR, and the results are reported in Table 6.

For overall and eastern China, the coefficients of interaction terms $T * P$ are 0.11 and 0.139, respectively, while for central and western China, the coefficient is not significant, indicating that the construction of HSR has an obvious significant role in promoting the economy of eastern China. Eastern China has a high population density and a large passenger demand. Thus, the opening of a HSR can effectively accelerate the flow of the various production factors among the regions and contribute to regional economic growth. However, due to the low population density in central and western China, the HSR's operational revenue can barely cover its operational cost, and the ticket price is comparatively expensive for citizens with low disposable income in the central and western regions. Therefore, HSR has no significant impact on the regional economy for central and western China.

For overall and eastern China, the coefficients of $T * P * C_1$ are 0.013 and 0.020, which are insignificant for the central and western regions. After the introduction of HSR, the influence of $C_1$ on the regional economy significantly decreased, indicating that airlines must adjust their air passenger networks to adapt to or avoid HSR competition. Since air cargo mainly uses the belly capacity of passenger aircraft, airlines cannot provide the market with high-quality air cargo services, and, therefore, cannot generate a remarkable impact on the regional economy.

Similarly, for China overall and eastern China, the coefficients of the interaction terms $T * P * C_2$ are 0.021 and 0.023, respectively, while the coefficient is not significant for central and western China. Compared with the regression results in Table 3, the influence magnitude of the international air cargo network clustering coefficient on the regional economy is significantly improved. Eastern China used to have dense air routes, but after the introduction of HSR, the airlines cut off some of their short- and medium-haul flights, and increased their long-distance flights, including both domestic and international flights. In short, the airlines adjusted their air passenger and related air cargo networks. As a consequence, the expansion of the international air cargo network improves the value of $C_2$ and the connectivity of the airports, creates comparative advantages for regions [56], enhances regional attraction [57], promotes employment [58], stimulates regional investment, and drives the development of local financial, advertising, tourism, and other service industries. Thus, the international air cargo network plays an even greater role in promoting the regional economy [59–61] after the introduction of HSR.

For other control variables, the proportion of the secondary industry production value to the overall GDP plays a significantly positive role on the GDP per capita for China overall, and eastern, central, and western China. Real fixed asset investments have a positive effect on the GDP per capita for China overall, and central and western China, but the impact for the eastern region is insignificant. R&D expenditures have a positive effect on the GDP per capita for both China overall and eastern China, but no effect on central or western China. Neither FDI nor the number of employees has a significant impact on the GDP per capita for China overall, or the different regions.

**Table 6.** Regression results for cities in different regions with the DID method.

| Variable | Overall | | | Eastern China | | | Central China | | | Western China | | |
|---|---|---|---|---|---|---|---|---|---|---|---|---|
| | (1) | (2) | (3) | (4) | (5) | (6) | (7) | (8) | (9) | (10) | (11) | (12) |
| $T*P$ | 0.11 (0.036) *** | 0.076 (0.031) | 0.066 (0.028) | 0.139 (0.049) *** | 0.101 (0.040) | 0.062 (0.040) | 0.003 (0.048) | 0.020 (0.038) | −0.027 (0.036) | −0.031 (0.083) | −0.047 (0.075) | 0.074 (0.059) |
| $T*P*C_1$ | 0.01 (0.004) | 0.013 (0.003) *** | | 0.017 (0.005) | 0.020 (0.005) *** | | 0.007 (0.005) | 0.006 (0.005) | | −0.017 (0.010) | −0.009 (0.009) | |
| $T*P*C_2$ | 0.016 (0.006) * | | 0.021 (0.006) *** | 0.014 (0.009) * | | 0.023 (0.009) ** | −0.002 (0.008) | | −0.003 (0.008) | 0.029 (0.014) | | 0.016 (0.012) |
| FDI | 0.003 (0.007) | 0.004 (0.007) | 0.004 (0.007) | −0.016 (0.012) | −0.014 (0.124) | −0.016 (0.013) | 0.012 (0.014) | 0.012 (0.013) | 0.012 (0.013) | 0.005 (0.010) | 0.003 (0.010) | 0.004 (0.010) |
| SEC | 0.319 (0.033) *** | 0.313 (0.033) *** | 0.308 (0.032) *** | 0.319 (0.109) *** | 0.649 (0.109) | 0.704 (0.111) *** | 0.895 (0.133) *** | 0.904 (0.131) *** | 0.832 (0.119) *** | 0.246 (0.040) *** | 0.249 (0.040) *** | 0.274 (0.037) *** |
| LAB | 0.002 (0.003) | 0.002 (0.003) | 0.002 (0.004) | 0.002 (0.005) | −0.003 (0.006) | −0.002 (0.006) | 0.004 (0.005) | 0.005 (0.005) | 0.006 (0.005) | −0.002 (0.008) | −0.001 (0.008) | −0.001 (0.008) |
| K | 0.100 (0.017) *** | 0.095 (0.017) *** | 0.100 (0.017) *** | 0.039 (0.025) | 0.033 (0.024) | 0.039 (0.025) | −0.124 (0.037) *** | −0.116 (0.034) *** | −0.131 (0.035) *** | 0.178 (0.028) *** | 0.177 (0.028) *** | 0.177 (0.028) *** |
| R&D | 0.084 (0.027) ** | 0.084 (0.027) ** | 0.083 (0.027) ** | 0.174 (0.047) *** | 0.171 (0.047) *** | 0.182 (0.048) *** | −0.035 (0.030) | −0.034 (0.030) | −0.035 (0.030) | 0.037 (0.044) | 0.047 (0.044) | 0.041 (0.044) |
| $R^2$ | 0.953 | 0.952 | 0.952 | 0.953 | 0.953 | 0.952 | 0.991 | 0.991 | 0.990 | 0.950 | 0.950 | 0.949 |

Note: $t$ stands for the variable of Treated, and P stands for Period. Note: * stands for significant at the 10% level, ** stands for significant at the 5% level, and *** stands for significant at the 1% level.

### 4.2.2. Impacts of the Air Cargo Network on Cities of Different Scales

The DID regression results for cities with different economic scales are reported in Table 7.

**Table 7.** Regression results for cities of different scales with the DID method.

| Variable | Overall | | | Super-Large Cities | | | Other Cities | | |
|---|---|---|---|---|---|---|---|---|---|
| | **(1)** | **(2)** | **(3)** | **(4)** | **(5)** | **(6)** | **(7)** | **(8)** | **(9)** |
| $T*P$ | 0.11 (0.036) *** | 0.076 (0.031) | 0.066 (0.028) | −0.354 (1.853) | 0.413 (0.345) | −0.354 (1.853) | 0.100 (0.036) ** | 0.049 (0.031) | 0.040 (0.036) ** |
| $T*P*C_1$ | 0.01 (0.004) | 0.013 (0.003) *** | | 0.052 (0.059) | 0.058 (0.057) | | 0.009 (0.004) | 0.013 (0.004) *** | |
| $T*P*C_2$ | 0.016 (0.006) * | | 0.021 (0.006) *** | −0.054 (0.203) | | 0.051 (0.203) | 0.014 (0.006) | | 0.020 (0.006) *** |
| FDI | 0.003 (0.007) | 0.004 (0.007) | 0.004 (0.007) | 0.043 (0.028) | 0.041 (0.027) | 0.037 (0.028) | 0.0004 (0.007) | 0.001 (0.007) | 0.001 (0.007) |
| SEC | 0.319 (0.033) *** | 0.313 (0.033) *** | 0.308 (0.032) *** | 0.375 (0.248) | 0.371 (0.246) | 0.338 (0.248) | 0.327 (0.032) *** | 0.318 (0.032) *** | 0.312 (0.032) *** |
| LAB | 0.002 (0.003) | 0.002 (0.003) | 0.002 (0.004) | −0.0001 (0.010) | −0.0002 (0.010) | 0.002 (0.010) | 0.002 (0.004) | 0.002 (0.004) | 0.002 (0.004) |
| K | 0.100 (0.017) *** | 0.095 (0.017) *** | 0.100 (0.017) *** | 0.068 (0.063) | 0.069 (0.062) | 0.066 (0.063) | 0.122 (0.018) *** | 0.112 (0.018) *** | 0.118 (0.018) *** |
| R&D | 0.084 (0.027) ** | 0.084 (0.027) ** | 0.083 (0.027) ** | 0.031 (0.066) | 0.031 (0.066) | 0.038 (0.066) | 0.109 (0.030) *** | 0.107 (0.030) *** | 0.107 (0.030) *** |
| $R^2$ | 0.953 | 0.952 | 0.952 | 0.944 | 0.930 | 0.929 | 0.957 | 0.956 | 0.956 |

Note: * stands for significant at the 10% level, ** stands for significant at the 5% level, and *** stands for significant at the 1% level.

For super-large cities, the coefficient of $T*P$ is insignificant, and it is 0.1 for the other cities. It is understandable for the coefficient to be positive for other cities. For the insignificant coefficient of super-large cities, even HSR can produce an agglomeration effect [62]. However, for super-large cities, the negative crowding effect may overwhelm the positive agglomeration effect, meaning that the introduction of HSR has no significant impact on the regional economy. For super-large cities, the effects of $T*P*C_1$ and $T*P*C_2$ are not significant, possibly because the existing railway (including the conventional railway and HSR) and air transport supplies are already sufficient to support economic development, while the newly added HSR's marginal contribution is negligible. For the other cities, the coefficients of $T*P*C_1$ and $T*P*C_2$ are 0.013 and 0.020, respectively, indicating that the international air cargo network has a higher impact on the regional economy than the domestic air cargo network does. For super-large cities, the GDP per capita is not affected by the proportion of the secondary industry production value to the overall GDP, the real fixed asset investments, or the R&D expenditure. However, for the other cities, all of the control variables play significant roles in promoting regional economic growth.

## 5. Conclusions and Policy Implications

With the rapid development of the transportation industry, air cargo transport plays an even more important role in facilitating regional economic growth. Under the impact of HSR, which has gradually evolved into the stage of network operations, China's civil aviation industry has gradually adjusted its air network in order to better cope with intermodal competition and take advantage of potential cooperation with HSR.

The air cargo network, measured by the clustering coefficient, has different impacts on the regional economy for different regions and cities at different scales. For eastern and central China, the higher the clustering coefficient of the air cargo network is for a specific city, the more significant its promotion effect is on the GDP per capita. Meanwhile, eastern China benefits the most from improvements to the air cargo network. However, the clustering coefficient of the international air cargo network has a negligible effect on the GDP per capita, because China's air cargo network is more domestically, rather than internationally, oriented. For most cities, the clustering coefficients of the domestic and

international air cargo network play a positive role. However, for super-large cities, the coefficient is insignificant, which is likely to be because the air cargo routes frequently overlap with the air passenger and HSR routes. Under the circumstances, when air passenger demands have priority over those of air cargo, and few space and time slot resources are allocated to passengers rather than cargo facilities in super-large cities, the impact of the air cargo network in super-large cities is rather small.

With the introduction of HSR in eastern China, the higher the clustering coefficient of the air cargo network is for a specific city, the more significant its promotion effect on the GDP per capita. However, there is no such effect for central or western China, mainly because HSR is too expensive for the citizens in central or western China to afford. Meanwhile, the air cargo coefficient of the air cargo network does not have a significant impact on the GDP per capita for super-large cities, but plays a significant role in promoting the GDP for other cities, mainly because there is a crowding effect for super-large cities. The corresponding policy implications are as follows.

Firstly, policy makers should develop favorable policies to strengthen the development of air cargo networks. The theoretical and empirical results show that both domestic and international air cargo networks can generate a positive impact on regional economic growth. Against the background of the COVID-19 pandemic, air passenger demand plummeted, and the demand for a more flexible, adaptable, and unified transport and logistics system increased. It has become imperative for the Chinese civil aviation industry to change its current air passenger priority strategy and address the importance of air cargo and logistics development. Scarce transportation resources, such time slots, and logistics facilities, etc., should be devoted to the development of air cargo networks. Policies should be implemented in order to encourage dedicated air cargo airport and service development.

Secondly, policy makers should encourage the full usage of HSR and civil aviation supply capacity to satisfy domestic and international air cargo demands. The gradual expansion of HSR in China has had a huge impact on the civil aviation industry, and both challenges and opportunities coexist. On the one hand, Chinese airlines have to adjust their network layouts and compete with HSR by developing medium- to long-distance air cargo products. On the other hand, airlines can cooperate with HSR to provide multi-modal express cargo services. Particularly, for the considerations of the 'Belt and Road' strategy and environmental protection, the Chinese government encourages rail–air intermodal cooperation, which provides further development opportunities for HSR and airlines. We should, therefore, synthesize the advantages of the air cargo network and the HSR distribution network to achieve a seamless connection, and thus provide the market with an innovative business model: an air–rail combined transport model.

Lastly, policy makers should cultivate and promote the development of market intermediaries. Market intermediaries include third-party logistics companies, cargo express companies, and forwarders, etc., who can improve air cargo service quality and efficiency, and can better respond to changes in the market demand. Preferential policies should be developed to encourage transnational corporations, such as FedEx, UPS, and DHL, and domestic transportation and logistics companies, such as Shunfeng, Jingdong, and Cainiao, etc., to cluster in or around airports in order to better integrate air cargo services and achieve the relevant economies of scale and scope.

**Author Contributions:** All of the authors participated in the discussion of the research framework and manuscript writing work. L.H. collected the data for the empirical study, calculated the clustering coefficient, and wrote part of the literature review; H.L. worked on part of analytical framework and did much of the English writing; N.Z. designed the overall structure and wrote the policy implications section; J.S. processed the data by applying the OLS and DID methods; X.L. assisted H.L. in writing part of the analytical framework; X.G. helped with the empirical results analysis. All authors have read and agreed to the published version of the manuscript.

**Funding:** This work was supported by the National Social Science Foundation of China under Grant Research on the Space-time Economy of Transport Big Data (Number: 19FJYB042).

**Conflicts of Interest:** The authors declare no conflict of interest.

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
