# Peer review of "The Influence of the Air Cargo Network on the Regional Economy under the Impact of High-Speed Rail in China"

_sustainability, doi:10.3390/su12198120_

Round 1

Reviewer 1 Report

Dear authors,

I find the research topic interesting, but I do not think the paper is publishable in its current form. There are some flaws both content- and form-wise which I will describe in the following.

Content:

  • It is not clear that's the goal of the paper. The abstract does not help, since it is not an abstract. It is a mix between an abstract, an introduction, and conclusions. As a matter of fact, it starts like the great majority of introductions of papers dealing with air cargo (mentioning the 35% value transported).
  • the methodology is not clearly stated. The OLS method is not introduced, while I do not think the DID method is ever mentioned explicitly. As example, I was not familiar with this method, and I would have appreciated the full term before the acronym (the first time it is mentioned of course)
  • it is also not clear how the rail and airport network are computed, and what their role is. How many airports are considered? Are frequencies between airports used as a proxy to assess the relevance of the connection?
  • I do not necessarily agree that high-speed trains are much more punctual, on average, than aircraft. If so, a reference or some objective data should be provided
  • results should be better described and interpreted. I tried to find an explanation in the text (which might be there, hence it might be my fault), but I do not understand why in every table the same element is repeated 3 times (e.g., overall, Eastern China, Central China)

I do believe all the aforementioned points should be addressed before having a paper that is ready for publication.

Form:

The clarity of the paper is further jeopardized by its form, which is rejection-quality right now. Apart from a slight re-structuring of the paper (e.g., a much better abstract, a better description of the dataset, etc), the full paper needs to be revised because the English level is very poor. I would suggest the authors just to re-read the "Author Contribution" section, to get an idea of the English level of the manuscript. Half of the sentences there are not in correct English. Some other points I want to highlight are:

  • line 70: "have the following specific paths of impacts" ---> not clear what is meant
  • lines 117-127: all this paragraph is not clear, and actually quite confusing
  • lines 146-149: you are re-stating the 35% of value transported, which is a repetition w.r.t. the abstract. This testifies even more that the paragraph should be removed from your revised abstract
  • many acronyms are being used without being defined (DID, SP, etc). This is not correct
  • lines 194-196: "after the operating" is not correct
  • line 205: "with certain results delivered" ---> not good academic English
  • line 237: instead of a generic reference on complex network theory, I think a more specific reference on scale-free properties of the Chinese air freight network should be provided
  • Eq.5 + line 336: "the number of actually connected edges by critical points" is not clear. That is the number of edges among the neighbors of node i. Also, given the denominator, I assume you are considering an undirected network, which might not be appropriate for cargo, where triangular routes are very common
  • Table 1: what do you mean with "Observe"? And "Mix" should be Min

To summarize, I think you have a nice story to tell, but this current version of the manuscript needs a severe upgrade.

Reviewer 2 Report

The paper focuses on the analysis of the influence on the economy of the air cargo under the impact of high-speed rail based on the DID method. Background information is adequately covered. The manuscript is organized properly and maintaining a clear logical flow across different sections. The analysis is carried extensively on the China territory. Some minor modifications, as suggested below, can improve readability and the significance of the results:
1. Please explain the acronyms (i.e. DID) when they appear for the first time in the manuscript.
2. I suggest a sensitivity analysis of the results with respect to the variables used and defined in the section 3.2.

Round 2

Reviewer 1 Report

Dear authors,

I appreciate the extra efforts you made, but in my opinion the paper is still not publishable. I would take more time to improve the paper rather than provide a quick and partial review.

As it concerns the results, I still think they are not clearly presented. Tables 3/4/5/6 are pretty massive, but the description of the associated results is scarce and not well-defined,

I also still think the complex network theory part is not properly introduced and justified. How many airports are considered? How many connections? Plus, several times you mention "air cargo networks" (plural form). Are you considering a single air cargo network that is the superimposition of the different carriers you cited on page 4, or different air cargo networks? In addition, I still believe the definition of the clustering coefficient (lines 376-378) is not understandable and actually wrong.

Regarding the introduction, you provide an introduction of 4 pages (i.e., very long) that does not really help the reader understand what will come next and is somehow repetitive and not to-the-point.

I am still very unhappy with the form. The English was improved in some parts, but some other parts are really rejection-level. As mentioned in my previous review, just check the "Author contribution" section, which is full of typos. That is a good indicator of the English level of the paper. In addition, Chinese characters appear within brackets as citations. They should not be there I guess.

All in all, I think this paper is borderline rejection, to be honest, in the current form. I know this might not please the authors, but I do believe consistent improvements are needed

Round 3

Reviewer 1 Report

I still think this sentence makes no sense in English (clustering coefficient section):

In which, Ci is the ratio of Ei (the number of actually connected edges (actually airline) by critical points (critical airport )) to the number of the maximum variables possibly connected.

Please revise
